# Plant-Derived Xanthones against Clostridial Enteric Infections

**DOI:** 10.3390/antibiotics12020232

**Published:** 2023-01-21

**Authors:** Ying Liu, Jianfei Zhu, Shaoqi Qu, Jianzhong Shen, Kui Zhu

**Affiliations:** 1National Key Laboratory of Veterinary Public Health Security, College of Veterinary Medicine, China Agricultural University, Beijing 100193, China; 2Guangdong Laboratory for Lingnan Modern Agriculture, Guangzhou 510642, China

**Keywords:** antibiotic substitutes, *α*-mangostin, *Clostridium perfringens*, necrotizing enteritis

## Abstract

Intestinal bacterial infections are a major threat to human and animal health. In this study, we found plant-derived antibacterial xanthones, particularly *α*-mangostin (AMG) from the mangosteen peel, exhibiting extraordinary activities against *Clostridium perfringens*. Structure–activity relationship analysis showed that prenylation modulated the activity of xanthones. The efficacy of AMG (4, 8, 20 mg/kg body weight) was also demonstrated in the broiler chicken necrotic enteritis model infected with *Clostridium perfringens.* In the models (n = 6 per group), feed supplementation of AMG maintained the homeostasis of the gut microbiome by reducing the colonization of clostridia and promoting the integrity of intestinal barriers via the upregulation of mucin expression. These results suggest that plant-derived xanthones may be a potential alternative to antibiotics for treating clostridial enteric infections in the clinic.

## 1. Introduction

Intestinal infections caused by pathogenic bacteria, particularly *Clostridium difficile* and *Clostridium perfringens*, are a serious health problem globally [1,2,3,4]. *C. perfringens* is the third common foodborne bacterial pathogen causing gas gangrene and food poisoning in humans. In the livestock and poultry industry, *C. perfringens* induces necrotic enteritis, a major concern which has led to severe economic losses [5]. Antibacterial compounds are still the mainstay of treatment for *C. perfringens* infections. However, inappropriate and irrational antibiotic treatments not only lead to recurrent infections and antibiotic-associated diarrhea, but also contribute to the emergence and spread of antibiotic resistance [6,7]. Thus, the discovery and development of antibiotic substitutes are urgently needed.

Different antibacterial strategies including fecal microbiota transplant (FMT) and probiotics have been employed to prevent clostridial enteric infections [8,9,10]. However, the risk of FMT and probiotics, regarding their possibility to transmit antibiotic-resistant bacteria, has caused much concern in clinic recently [11,12]. Moreover, the unclear mechanism of probiotics limits its clinical application [13,14]. Compared to these alternatives, plant-derived natural products with antibacterial activity possess robust advances in accessibility, structural diversity, and efficacy. Bioactive molecules extracted from plants have shown great potential for the development of antibiotic substitutes against various bacteria [15,16,17,18].

Plants are abundant in a diverse group of flavonoids with robust activity against infection, inflammation, and tumors. The high hydrophobicity characteristic of flavonoids has indicated the potential for oral medication for intestinal infections to have low bioavailability [19,20,21]. However, it has been shown that flavone and flavonol could be metabolized and degraded by enteric *Clostridium orbiscindens* [22], paralyzing or abolishing their antibacterial efficacy in the gut (Figure 1). Notably, one group of flavonoids, xanthones, has been claimed to be resistant to the degradation of gut microbiota, owing to its unique tricyclic privileged structures [23,24]. Xanthones are also promising membrane-disrupting agents against bacterial infections, without detectable resistance. Thus, plant-derived xanthones are an alternative for treating intestinal infections with *Clostridium perfringens*.

Recently, we found that xanthones, especially α-mangostin extracted from mangosteen peels, exhibited robust antibacterial activities against methicillin-resistant *Staphylococcus aureus* (MRSA) and vancomycin-resistant enterococci (VRE) [18,25]. Given the attractive efficacy, we evaluated the potential of plant-derived xanthones as an alternative strategy for the treatment of enteric infections using the broiler chicken necrotic enteritis model. Our findings provide an alternative strategy for treating clostridial enteric infections in human beings and animals.

## 2. Results

### 2.1. Structure–Activity Relationship of Xanthones

We previously found the lead compound α-mangostin (AMG), from five categories of flavonoids, to be a promising antibacterial agent with minimal inhibitory concentrations (MIC_50_) of 0.5 μg/mL against MRSA and VRE. Herein, we further evaluated the antibacterial activity of AMG and other types of flavonoids against *C. perfringens*. Compared to mulberrin (flavonoid), glabrol (flavanone), lupalbigenin (isoflavone), and isobavachalcone (chalcone), AMG showed better antibacterial activity against five genotypes of *C. perfringens*, with MIC values of 0.5 μg/mL (Table 1). Meanwhile, we found that AMG exhibited superior activities to routinely used antibiotics including tylosin, tilmicosin, and lincomycin to treat *C. perfringens* in livestock and poultry (Appendix A). To further test the generality of AMG, we expanded to 79 clinical *C. perfringens* isolates and found that the MIC_50_ and MIC_90_ of AMG were kept at 0.5 μg/mL, indicating much more robust efficacy than tylosin with the MIC_50_ of 1 μg/mL and MIC_90_ of 4 μg/mL (Appendix A, Appendix A). In addition, AMG exhibited the same antibacterial activity under anaerobic conditions against different bacteria including MRSA and VRE (Appendix A), suggesting that AMG may be efficient against enteric infections *in vivo*.

Subsequently, we examined the structure–activity relationship of AMG against *C. perfringens*. Generally, we found that the position of prenylation simultaneously at C2 and C8 of the xanthone skeleton, such as α-, β-, and γ-mangostin, is critical for antibacterial activity, consistent with previous studies [26,27]. Meanwhile, the hydroxyl group at C3 on the parent ring structure promoted antibacterial activity, whereas the methoxyl group at C7 inhibited the efficacy (Figure 2, Appendix A). In addition, AMG displays many desirable pharmacological properties, such as low molecular weight, six hydrogen acceptors, and five rotatable bonds (Appendix A), according to Lipinski’s rules and eNTRy rules [28]. Nevertheless, AMG suffers from high lipophilicity (log P = 4.64) and low aqueous solubility, probably due to the tetracyclic rigid plane composed of an intramolecular hydrogen bond between the hydroxyl group at C1 with the carbonyl group. Taken together, AMG is a potential lead against clostridia, particularly for the treatment of intestinal tract-associated infections.

### 2.2. Stability and Safety Assessment of AMG

Given that many herbal substances or preparations are unstable and toxic for therapeutic purposes [29,30,31], we first investigated the stability of AMG under environmental conditions. We found that AMG showed a two- to four-fold increase in MICs against antibiotic susceptible *S. aureus* and MRSA under light conditions (Figure 3A). Meanwhile, AMG maintained the antibacterial activity under pretreatment with different temperatures ranging from −20 °C to 37 °C for 1 h. However, heat treatments showed reduced trends of antibacterial activity, indicated in Figure 3B. Acid resistance is an important feature of oral drugs. Markedly, AMG exhibited a high tolerance to acid (pH = 1, Figure 3C). Compared to the physiological pH condition, the structure of AMG was consistently maintained after being incubated under physiological pH ranging from 3.8 to 7.4 based on the UV absorption spectrum (Figure 3D). In addition, the resistance to microbial degradation is a prerequisite for AMG against intestinal infections. The widely distributed *Eubacterium* spp. and *Flavonifractor* spp. were dominant species in the cecum (top 30) in 112 chicken and 20 swine health samples (Figure 3E). Although these microbes can produce flavone reductase (FLR) to specifically catalyze flavones and flavonols, they are not able to degrade xanthones. Correspondingly, AMG displayed comparable antibacterial activity after being incubated in simulated gastric and intestinal fluids (Appendix A). Lastly, we assessed the acute toxicity of AMG and observed a median lethal dose (LD_50_) of more than 1000 mg/kg in either SD rats or ICR mice. It is in agreement with the previous report that the LD_50_ value of mangosteen extract is an approximate of 1000 mg/kg in BALB/c mice [32]. Therefore, AMG is a low-toxicity substance according to the acute toxicity-grading standard for oral compounds. Collectively, our findings indicate that AMG is a promising alternative to antibiotics for the treatment of gastrointestinal infections.

### 2.3. AMG Modulates the Composition of Intestinal Microbial Community

Given the attractive efficacy of AMG, we further investigated its therapeutic potential against intestinal infections in vivo. According to a previous method, the model of necrotizing enteritis was established in *Arbor Acres* broiler chickens [33]. At 14 days of age, broilers were orally, once per day, given *C. perfringens* suspension for five days (Figure 4A). The pathological autopsy of the challenged broilers showed obvious flatulence in the intestinal tract, especially in the posterior segment of the small intestine. Meanwhile, the bacterial load of *C. perfringens* increased significantly in the cecum of infected broilers (Appendix A). These results indicated that the necrotizing enteritis model was successfully established.

The gut microbial community after antibiotic treatment may provide a favorable environment for *C. perfringens*, and cause antibiotic-associated diarrhea [34,35,36]. To verify the potential effect of AMG on intestinal microbiota, we supplemented the diet with AMG for three consecutive days at 24 h after infection. Cecal contents were subsequently collected, and microbiota was analyzed based on 16S rRNA gene sequencing. Metagenomics analysis indicated that Firmicutes were the predominant phylum in the cecum (Appendix A). The increased number of beneficial bacteria, such as *Lactobacillus* spp., and reduction of pathogenic bacteria, such as *Brucella* spp., was demonstrated in AMG treatment groups (Figure 4B). In addition, the *α*-diversity analysis showed that species abundance and diversity were maintained after AMG treatment, compared to the uninfected groups (Figure 4C and Appendix A). These findings are consistent with the operational taxonomic units (OTUs) result (Figure 4D). In addition, the *β*-diversity analysis showed that the composition of the intestinal microbial community after AMG treatment is similar to that in uninfected groups (Figure 4E and Appendix A). Last, based on LEfSe analysis, we found that *Bacillus* spp. dominate the cecum flora in the AMG treatment group, whereas *Clostridium* spp. dramatically increased in infections. Similarly, the LDA scores indicated that the relative abundances of *Blautia* and Lachnospiraceae were significantly higher in the challenged group than the AMG treatment group (Figure 4F and Appendix A). AMG sharply inhibited pathogenic bacteria and reversed the dominant bacteria as the control. These results suggest that AMG reduces the abundance of pathogenic bacteria and potentiates the growth of probiotics, especially *Lactobacillus*, thereby efficiently ameliorating the dysbiosis in necrotizing enteritis.

### 2.4. AMG Effectively Controls Clostridial Enteric Infections

Modulation of the host immune response is also an effective method to control necrotic enteritis with *C. Perfringens.* To further evaluate the effect of AMG on the host, we first assessed the expression levels of inflammatory factors in mammalian cells after AMG treatment based on quantitative PCR (qPCR) and ELISA tests. We found that AMG inhibited IL-1β expression, whereas it showed no effects on TNF-α (Figure 5A and Appendix A). Interestingly, AMG at the low level (1.2 μmol/L) enhanced the phagocytic activity of macrophages (Figure 5B). Therefore, AMG exhibits anti-inflammatory and immunomodulatory potential.

The therapeutic effect of AMG in vivo was subsequently evaluated using the necrotizing enteritis model. AMG had a significant inhibitory effect on *C. perfringens* in a dose-dependent manner (Figure 5C). Given that *C. perfringens* usually colonizes in the cecum, we further evaluated the distribution of AMG and found that AMG was predominantly found in the cecum as well (Figure 5D and Appendix A). Moreover, the intestinal villi structure of broilers in the infected groups disintegrated entirely based on pathological analysis. Inspiringly, AMG treatment almost maintained the intestinal villi structure of broilers indicating that AMG effectively alleviates the pathological lesion of intestinal tissue caused by necrotizing enteritis (Figure 5E). Enteric clostridial infection has been reported to reduce the expression of intestinal tight junction genes, and promotes the hydrolysis of tight junction proteins [37]. We found that AMG significantly up-regulated the expression of mucin protein MUC2 through qPCR analysis (Figure 5F). This result was also confirmed by immunofluorescence colocalization of MUC2 (Figure 5G). Thus, AMG is effective against necrotizing enteritis in chickens, suggesting that plant-derived molecules are prospective sources for developing antibiotic alternatives to prevent gastrointestinal infections.

## 3. Discussion

Anaerobic *Clostridium* spp., especially *Clostridium difficile* and *Clostrdium perfringens*, ubiquitously distribute in the abiotic environment and biotic intestinal tracts. These pathogens are the primary causative agents of intestinal diseases such as necrotizing enteritis and antibiotic-associated diarrhea [38,39]. Antibiotics are routinely used to control broiler necrotic enteritis such as lincomycin, tylosin, penicillin, and tilmicosin. However, antibiotic resistance and antibiotic-associated diarrhea become a serious problem in the treatment of *C. perfringens* [40]. Plant-derived compounds have drawn much attention as a promising source of antibiotic substitutes [41,42,43] due to their abundance, versatile chemodiversity, and high accessibility [44]. Moreover, the unique membrane targeting mechanism without antibiotic resistance indicates the antibacterial potential [18]. To find a novel strategy to control necrotic enteritis infected with *C. perfringens*, we detected the antibacterial activity of plant-derived flavonoids against *C. perfringens*. The lead compound AMG of xanthones exhibits robust bactericidal activity against *C. perfringens*, with MIC_50_ of 0.5 μg/mL and MIC_90_ of 0.5 μg/mL. AMG could not only maintain the homeostasis of the gut microbiome by reducing the colonization of clostridia, but could also promote the integrity of intestinal barriers via the upregulation of mucin expression. Thus, plant-derived xanthones provide an alternative strategy to treat clostridial enteric infections.

The therapeutic efficacy frequently suffers from pharmacokinetic behaviors. First, pH stability is the unique advantage of gastrointestinal drugs. Inspiringly, AMG, the candidate of xanthones, maintained the structure and the antibacterial activity after being incubated at pH = 1 for 1 h. Moreover, previous studies have demonstrated the poor bioavailability of AMG, suggesting its low gastrointestinal absorption [45,46]. We found AMG was mainly distributed in the cecum (Figure 5D), supporting the idea that AMG was hardly absorbed in the gastrointestinal tract. Most importantly, it has been confirmed AMG could not be metabolized and degraded by enteric *C. orbiscindens*, owing to the unique tricyclic privileged structures [22] (Figure 1). Taken together, AMG can exert its efficacy fully in the intestine, reducing the dosage and administration times and decreasing the risk of drug resistance in a superior manner to other antibiotics.

Bioactive substances with dual functions of antibacterial and immune regulation are potential choices for antibacterial therapy [47]. Encouragingly, AMG exhibits anti-inflammatory and immunomodulatory potential (Figure 5A,B) and effectively alleviate the pathological lesions of intestinal tissue (Figure 5E,F). Bacteria and host-oriented dual-functional antibacterial xanthones, including AMG, provide an effective strategy for the treatment of intestinal infection with *C. perfringens* (Figure 6).

## 4. Materials and Methods

### 4.1. Chemicals and Bacterial Strains

Standard compounds of xanthone were purchased from Chengdu Biopurify Phytochemicals Ltd. with a purity of ≥98%. Before starting the experiment, compound stock solutions were prepared in dimethylsulfoxide (DMSO, Aldrich, Taufkirchen, Germany) at a concentration of 5120 μg/mL. *S. aureus* ATCC 29,213 was obtained from the American Type Culture Collection (ATCC); CVCC2030 and CVCC60102 were purchased from China Veterinary Microbial Species Preservation and Management Center, China Veterinary Drug Control Institute. Other *C. perfringens* clinical isolates were obtained from ileum contents, cloacal swabs, or feces of chickens, pigs, and cattle from large-scale livestock and poultry farms during 2019 in China, then identified and preserved by the National Center for Veterinary Drug Safety Evaluation at China Agricultural University, as noted in Table 1.

### 4.2. Antibacterial Tests

Minimum inhibitory concentrations (MICs) were determined by the broth microdilution method following the Clinical and Laboratory Standards Institute (CLSI) 2022 guidelines. The range of concentrations typically employed for each experiment was 128–0.25 μg/mL (2.56% final DMSO concentration). Tylosin was used as the positive control in this experiment. All *C. perfringens* bacterial strains were sub-cultured on Brain Heart Infusion agar (BHA, Bridge Technology Co., Ltd., Beijing, China) with 5% sheep blood and incubated overnight in an anaerobic jar containing 10% H_2_, 10% CO_2,_ and 80% N_2_ at 37 °C. The test medium for *C. perfringens* species was Fluid Thioglycollate Medium (FTG, Bridge Technology Co., Ltd., Beijing, China), and bacterial suspensions were diluted to obtain a final inoculum of 1.5 × 10^6^ CFU/mL for broth microdilution experiments. The lowest concentrations of compounds with no visible growth of bacteria were the MICs after incubation at 37 °C for 24 h.

### 4.3. Stability of AMG

To determine the light stability, AMG was pretreated with sunlight exposure for 3 h, and the residual antimicrobial activity was evaluated against *S. aureus*. To determine the thermal stability and pH stability, AMG was treated at a temperature ranging from −20 to 60 °C, and at pH ranging from 1.0 to 6.0 at 37 °C for 1 h, respectively. Subsequently, all samples were readjusted to pH 7.0 to determine the residual antimicrobial activity against *S. aureus*.

### 4.4. Antibacterial Activity of AMG in the Simulated Gastrointestinal Model

Simulated gastric fluid (SGF) and simulated intestinal fluid (SIF) were prepared using a commercial kit (Phygene, Fuzhou, China). AMG was first mixed with SGF. After 2 h incubation at 37 °C with continuous shaking at 100 rpm, the SGF was adjusted to neutral pH with 2 mol/L NaOH and then mixed with an equal volume of 2 × SIF. The mixtures were two-fold diluted in FTG Medium and mixed with *C. perfringens* suspended in BHA at a final concentration of 1 × 10^6^ CFUs mL^−1^ in 96-well plates. Subsequently, the mixtures were incubated anaerobically at 37 °C for 5 h to evaluate their antibacterial activity.

### 4.5. In vitro RT-qPCR and ELISA Assays

MH-S cells were seeded at 1 × 10^6^ cells per well in 12-well culture plates (Corning Incorporated Co., New York, NY, USA) and incubated at 37 °C overnight to form monolayers. MH-S cells were treated with 10 μmol/L AMG, and 1 μg/mL LPS was added and co-incubated for 4 h after half an hour. In addition, Dexamethasone (DXM) was used as a controlled drug. After that, supernatants of cell cultures were acquired by centrifugation at 1000× *g* for 5 min at 4 °C. The levels of TNF-α, IL-6, and IL-1β in the supernatants were detected by commercial ELISA kits (BioLegend, Inc., San Diego, CA, USA) following the manufacturer’s instructions. Meanwhile, Total RNA was extracted using a HiPure Total RNA Plus Mini Kit (Magen Biotechnology Co., Guangzhou, China), and 1 μg of extracted RNA was reverse transcribed with a PrimeScript™ RT reagent Kit (Takara, Dalian, China) following the manufacturer’s protocol. The messenger RNA levels of TNF-α, IL-6, and IL-1β relative to the control genes *β*-actin in MH-S cells were determined by real-time PCR tests using SYBR Green Master Mix. The primer sequences are listed in Appendix A.

### 4.6. Phagocytosis Assay

MH-S cells were seeded into a 12-well plate at a concentration of 3 × 10^5^ cells per well at 37 °C for 8 h. Cells were treated with 1.2 μmol/L AMG for 24 h, and the medium of the phagocytosis inhibition positive control group was replaced with DMEM containing 1 μmol/L Cytochalasin D (CCD) 1 h before the incubation process was completed. The spent culture medium at the end of the incubation process was removed and the MH-S cells washed with PBS once. We suspended the FITC-labeled particles into DMEM (1:1000 dilution) and added them to the plates (500 μL/well), this proceeded for half an hour, and then we removed the medium and washed with PBS again. We trypsinized cells and resuspended them with cold PBS to stop phagocytosis. Finally, the cell suspension was filtered through a 300-mesh copper grid and run in a flow cytometer to detect FITC fluorescence.

### 4.7. Animals and Ethics Statement

Commercial 1-day-old *Arbor Acres* (AA) broiler chickens were purchased from Beijing Arbor Acres Poultry Breeding Co., Ltd. (Beijing, China). All animal studies were carried out at China Agricultural University, and all the animals were randomized to cages for each experiment and had free access to food and water. The animals were maintained in strict accordance with the regulations of the Administration of Affairs Concerning Experimental Animals, approved by the State Council of the People’s Republic of China (14 November 1988) and by the China Agricultural University (CAU) Institutional Animal Care and Use Committee guidelines (ID: SKLAB-B-2010-003) approved by the Animal Welfare Committee of CAU.

### 4.8. In Vivo Necrotic Enteritis (NE) Disease Model

The NE disease induction model was performed as previously described [33,48]. Briefly, a total of 36 commercial 1-day-old Arbor Acres broiler chickens (Beijing Arbor Acres Poultry Breeding Co. Ltd., Beijing, China) were divided into four experimental groups, including a non-challenged control, a challenged control group, and two challenged groups supplemented with either AMG (4 mg/kg, 8 mg/kg or 20 mg/kg body weight) or amoxicillin (1 mg/kg body weight). From day 0 to day 11, all broiler chickens received a control diet with no antibiotics. On day 12 the feed was changed to a mixture of 50% wheat-based and 50% fish meals. On the evening of the 13th day, the feed was withdrawn, and each bird was orally challenged with 1.0 mL of *C. perfringens* culture (10^9^ CFU/mL). From day 18 to day 20, birds were treated with AMG for three days. On day 21, chickens were euthanized with inhaled carbon dioxide gas, and their small intestines (duodenum to ileum) were examined for gross necrotic lesions. *C. perfringens* was recovered from the intestines of killed birds by culturing on Tryptose Sulfite Cycloserine Agar Base (TSC, Bridge Technology Co., Ltd., Beijing, China).

### 4.9. Microbiota Analysis

Total genome DNA from about 150 mg of cecum content of broiler chickens treated with AMG was extracted using the CTAB/SDS method. The 16S rRNA gene V3-V4 region was amplified using specific primers F341 (CCTACGGGRSGCAGCAG) and R806 (GGACTACVVGGGTATCTAATC). The chimera sequences were filtered using USEARCH, and then the remaining sequences were assigned to generate the same operational taxonomic units (OTUs) at 97% similarity. The alpha analysis based on USEARCH alpha_div was conducted to reveal the diversity indices. The beta diversity analysis was performed using USEARCH beta_div. Linear discriminant analysis was performed with LefSe (version 1.0.8) with default parameters (LDA score ≥ 2). Anosim analysis was used, Vegan Package(Version 2.4.2). The data were analyzed using R version 3.4.3 and GraphPad PRISM 8 (GraphPad Software, San Diego, CA, USA). All data are expressed as the mean ± SD.

### 4.10. Tissue Distribution of AMG

The distributions of AMG in the cecum were conducted according to a previously reported method [45]. About 0.2 g of cecum contents was taken from each chicken and homogenized with 1 mL sterile water at 5000 rpm for 10 min. AMG was extracted by adding 2 mL ethyl acetate, followed by 5 min vortex and 5 min ultrasonic extraction. After centrifuging at 9000× *g* rpm at 4 °C for 10 min, the supernatant was collected for nitrogen blow drying at 40 °C and redissolved in 500 μL 80% acetonitrile. Finally, samples were centrifuged at 14,000× *g* rpm for 15 min, and the supernatant was collected into filter vials and stored until LC-MS/MS analysis.

### 4.11. Determination of AMG

The analytical method for measuring AMG was modified from a previous report [49]. The sample and standard compound were dissolved with acetonitrile and completed with 0.22 µm pore filter membranes. Subsequently, 5 μL of the supernatant was directly injected into a reversed-phase C18 column (Cadenza CD-C18, 2 mm × 75 mm i.d., 3-μm particle size; ImtaktKyoto, Japan) and kept at 4 °C until injection. The mass transitions for AMG were *m/z* 411.10/355.10 (collision energy, 21.0 eV), in the multiple reaction monitoring (MRM) mode with positive ionization. These analytes were separated on the column with a gradient elution consisting of 0.1% formic acid in acetonitrile (A phase) and 0.1% formic acid in water (B phase) at a flow rate of 0.30 mL/min. The gradient conditions were as follows: 0.00–1.00 min, 80% A; 1.00–4.00 min, 80–100% A; 4.00–5.00 min, 100% A; 5.00–5.50 min, 100–80% A; 5.50–6.50 min, 80% A for 1 min. All analytical data were processed using the Shimadzu lab solutions (Version 5.106).

### 4.12. Cecal Epithelial Integrity Assays

The RNA expression levels of cecal tight junction proteins Claudin-1, Occludin, and ZO-1 and mucin protein MUC2 were measured in each group. Briefly, about 1 cm of cecal tissue was cut into a precool centrifuge tube containing 1 mL Buffer RL on ice. We homogenized the tissue until no pieces could be seen, then centrifuged the homogenate at 14,000× *g* rpm at 4 °C for 5 min. After that, total RNA was extracted using a commercial RNA extraction kit.

### 4.13. Immunofluorescence Staining

Briefly, tissues were cleared with xylene, dehydrated in an ethanol series, then embedded in paraffin. Then, the paraffin section slide was dewaxed, and antigen retrieval was conducted by transferring the slide to a container with unmasking solution EDTA (pH = 9.0) and microwaving. After washing, the samples were incubated with Autofluorescence Quenching Kit for 5 min and rinsed with running water for 10 min. Subsequently, blocking solution BSA was added for 30 min. Then, samples were incubated with primary antibodies overnight at 4 °C in the humid box. After repeated washing, the samples were incubated with secondary antibodies at room temperature for 50 min protecting them from light. After repeated washing, DAPI was added for 10 min to visualize DNA. Images of the samples were obtained after the mounting step using confocal scanning microscopy. The fluorescence of DAPI (blue) was detected at the excitation wavelength of 380 nm and emission wavelength of 420 nm. The fluorescence of MUC2 (red) was detected at an excitation wavelength of 510~560 nm and an emission wavelength of 590 nm.

### 4.14. Statistical Analysis

Experimental results were analyzed for statistical significance using GraphPad Prism 8.0.1 (GraphPad Software Inc.). Differences were analyzed by Student’s *t*-test and one-way ANOVA. The values reported were shown as mean ± standard deviation (s.d.). *p* values were indicated in the figures, and not significant (*p* > 0.05) was marked with n.s.

## 5. Conclusions

Our results demonstrate the potential of xanthones against clostridial enteric infections, providing an alternative strategy for the treatment of clostridial pathogens-associated infections. Plant-derived xanthones may be a promising source for the discovery and development of novel antibiotics to treat clostridial enteric infections in clinic.

## Figures and Tables

**Figure 1 antibiotics-12-00232-f001:**
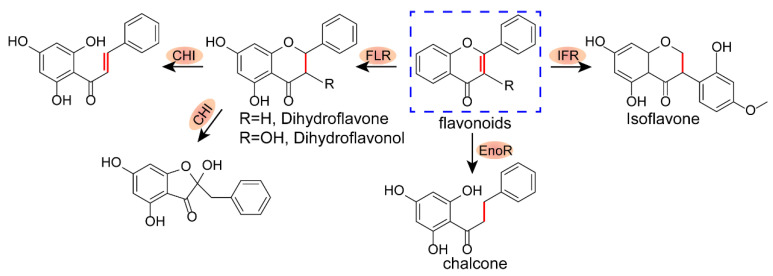
The proposed microbial metabolic pathway of flavonoids. The blue boxes mark the parent rings of flavonoids, and enzymes that metabolize flavonoids are shown (orange ellipses). FLR, flavone reductase; CHI, chalcone isomerase; EnoR, enoate reductase; IFR, isoflavone reductase.

**Figure 2 antibiotics-12-00232-f002:**
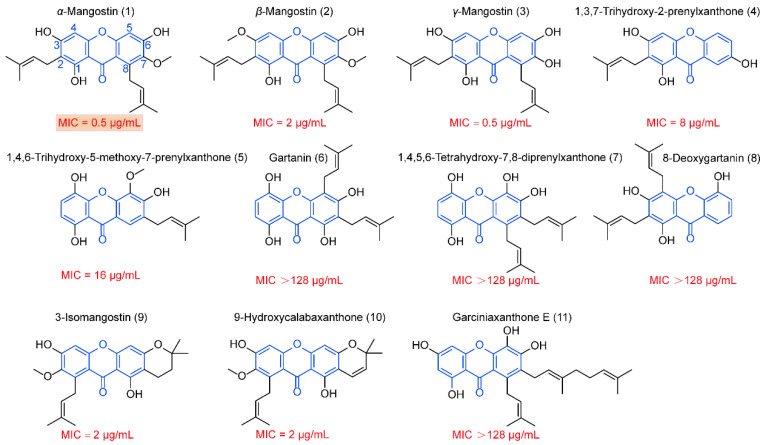
Structure–activity relationship analysis of xanthones. The parent structure of xanthones was marked in blue. MIC values of AMG and its analogs against *C. perfringens* are indicated below the corresponding structural formula.

**Figure 3 antibiotics-12-00232-f003:**
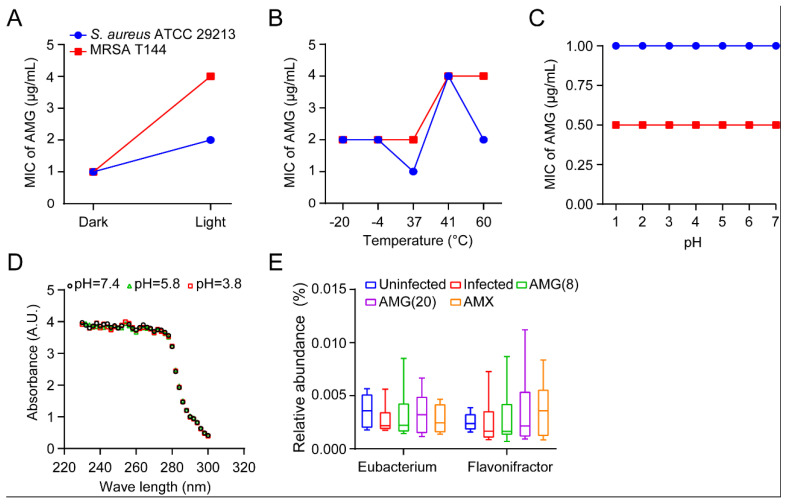
Stability of AMG and AMG activity against *C. perfringens* isolates. (**A**–**C**) Residual antibacterial activity of AMG after being treated at different illumination for 3 h (**A**), at different temperatures ranging from −20 °C to 60 °C for 1 h (**B**), and at different pH ranging from 1.0 to 6.0 at 37 °C for 1 h (**C**). *S. aureus* ATCC 29,213 and MRSA T144 were used as test strains. (**D**) The UV absorption spectrum of AMG in methyl alcohol under physiological pH. (**E**) The average relative abundance of *Eubacterium* spp. and *Flavonifractor* spp. in cecum samples were evaluated based on metagenomics sequencing of 16S rDNA, including the uninfected group, infected group, and treatment group (AMG or AMX). *N* = 6 broilers per group. Experiments in (**A**–**C**,**E**) were performed on two biological replicates.

**Figure 4 antibiotics-12-00232-f004:**
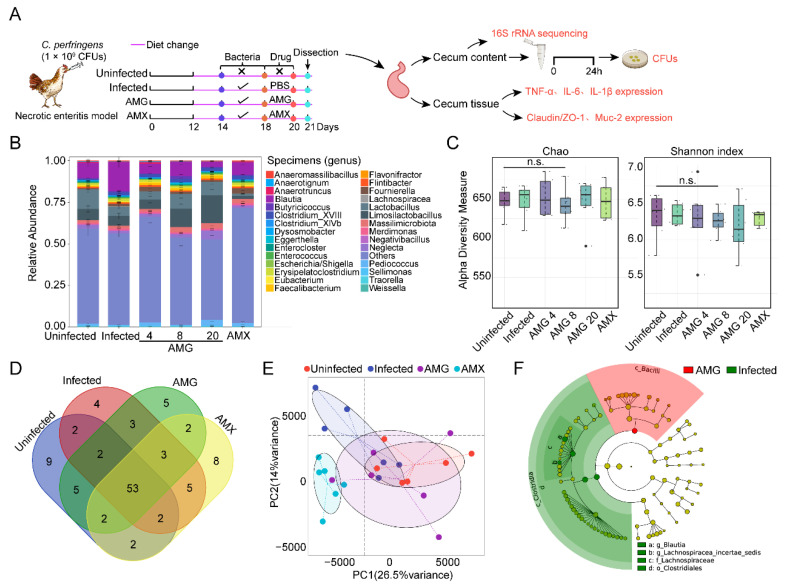
AMG modulates the composition of microbial community. (**A**) Scheme of the experimental protocol. *Arbor Acres* broiler chickens (6 birds per group) were infected with *C. perfringens* (1.0 × 10^9^ CFUs per bird) from day 14 to day 18, followed by treatments with AMG (4, 8, and 20 mg/kg body weight) for three consecutive days, and amoxicillin (AMX, 1 mg/kg body weight) was used as the positive control. Intestinal contents and tissue were then removed for subsequent analysis. (**B**) Relative abundance of different annotated species in the gut microbiota of broilers at genus levels (Top 30). (**C**) Alpha diversity in intestinal samples of broilers. (**D**) Venn diagram of the unique numbers of core ZOTU. (**E**) Principal component analysis (PCA) based on the unweighted UniFrac distance matrix, representing 40.5% of the total variation. (**F**) Phylum and genus differentially represented between infected (green) and AMG (red) samples. All data were presented as means ± SD (*n* = 6).

**Figure 5 antibiotics-12-00232-f005:**
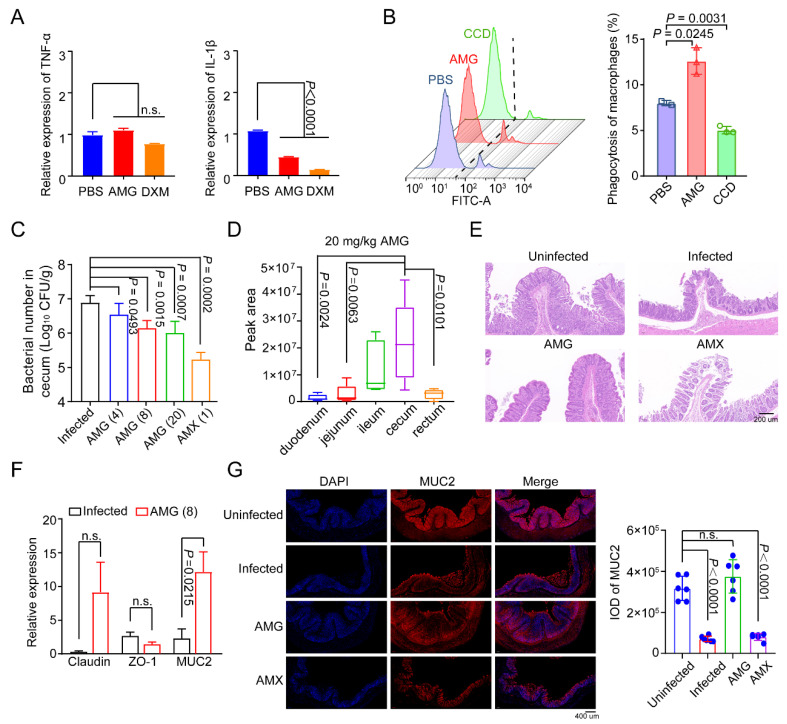
AMG is efficacious against necrotizing enteritis. (**A**) AMG effectively inhibited the expression of IL-1β in MH-S cells. MH-S cells were treated with 10 μmol/L AMG or dexamethasone (DXM) for half an hour, followed by 1 μg/mL LPS for 4 h. (**B**) Low level of AMG (1.2 μmol/L) enhanced the phagocytosis of macrophages. Cytochalasin D (CCD, 1 μmol/L) was used as a positive control. (**C**) Bacterial loads of *C. perfringens* in cecum after AMG treatments (4 mg/kg, 8 mg/kg, 20 mg/kg body weight) for three days. AMX was used as a positive control (*n* = 6). (**D**) The distribution of AMG (20 mg/kg body weight) in gut was analyzed by liquid chromatography-tandem mass spectrometry (LC-MS/MS) test after oral gavage in broilers. (**E**) AMG (20 mg/kg body weight) ameliorated the pathogenicity in broilers caused by *C. perfringens*. Scale bar = 200 μm. (**F**) AMG promoted the expression of MUC2 in cecum. Relative expression of claudin, ZO-1, and MUC-2 were determined by qRT-PCR. All data were presented as means ± SD. *p* values were determined by one-way ANOVA. (**G**) AMG increased the secretion of MUC2. Nuclei were stained with DAPI (blue), and expression of MUC2 (red) was detected by Cy-3 labeled goat anti-rabbit IgG. The integral optical density (IOD) values of positive cells were statistically analyzed. Scale bar = 400 μm.

**Figure 6 antibiotics-12-00232-f006:**
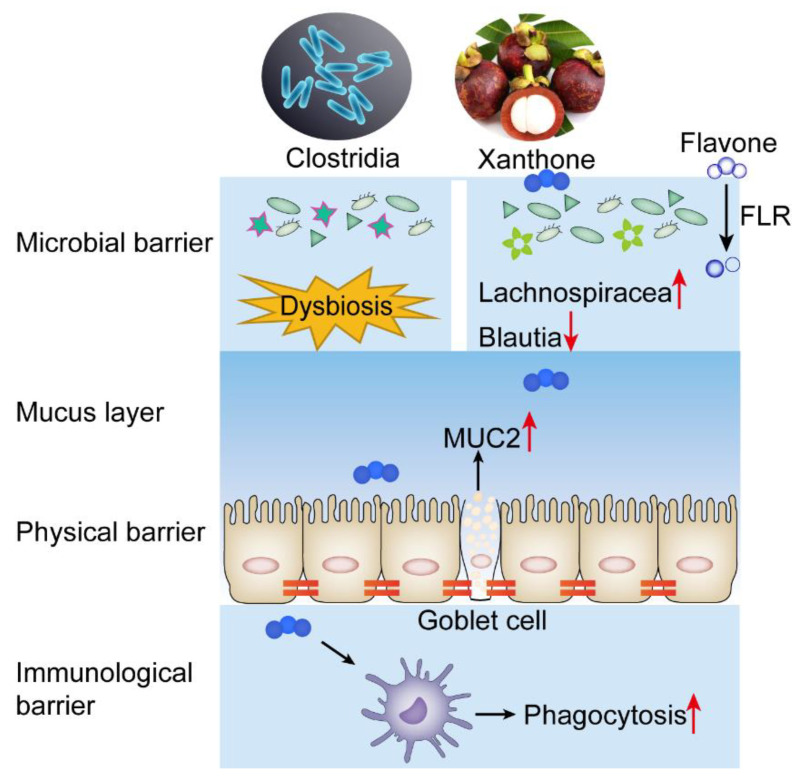
Scheme of plant-derived xanthones as potential antibiotic substitutes. The xanthones are resistant to the degradation of gut microbiota and can effectively modulate the composition of intestinal microbials, reduce bacterial load, and protect intestinal mucosa.

**Table 1 antibiotics-12-00232-t001:** Antibacterial activities of flavonoids against *C. perfringens*.

*C. perfringens*	Genotype	MIC (μg/mL)				
		AMG	Isobavachalcone	Mulberrin	Lupalbigenin	Glabrol
CVCC2030	A	0.5	2	2	0.5	1
CVCC60082	B	0.5	2	2	0.5	1
CVCC60101	C	0.5	2	1	0.5	1
CVCC60201	D	0.5	2	1	1	1
20SX 1RX187	A	0.5	4	1	1	1
20SJ BNP12	A	0.5	4	2	1	1
19NM 2CM8	A	0.5	4	2	0.5	2

## Data Availability

Not applicable.

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
