# Peer review of "Plant-Derived Xanthones against Clostridial Enteric Infections"

_antibiotics, 2023, doi:10.3390/antibiotics12020232_

Round 1

Reviewer 1 Report

The research article explained the role of xanthos molecule anti microbial functions to treat intestinal bacterial infection. The article came with nice piece of witness and provided sufficient results to publish.

1. The antibacterial activity was measured from -20 to 60 degree celcius, where the activity differed from 0 to degree celcius what might be the reason the acitivity did not show any difference below 0 degree.

2. It would be nice for readers to explain why author carried out at below zero degree celcius and how the experiment was performed.

3. AMG posses low MIC at 37 degree celcius what would be significant

4. It would be nessasary to explian the reason to conduct the activity from pH1.0 to  pH6.0 at the temperature where the AMG showed low MIC towards S.aureus. 

5. How the author carried out activity at dark also at below 0 degree celcius.

6. Why author did not show AMG activity at physilogical pH

7. Author needs to be improve the discussion part.

Author Response

Point-by-point response

Reviewer Comments:

Reviewer #1:

The research article explained the role of xanthones molecule antimicrobial functions to treat intestinal bacterial infection. The article came with a nice piece of witness and provided sufficient results to publish.

Response: We thank the reviewer for recognizing the novelty and significance of the work.

Comment 1: The antibacterial activity was measured from -20 to 60 degrees Celcius, where the activity differed from 0 to degree Celcius what might be the reason the activity did not show any difference below 0 degrees?

Response: We sincerely appreciate the valuable comment. To determine the thermal stability, AMG was treated at a temperature ranging from -20 to 60 °C for 1 h, then tested the antibacterial activity against S. aureus at 37 °C. Pretreatment at a different temperature only resulted in a 1-2 fold change in the MIC values, suggesting that AMG remains stable (including conditions below 0 degrees).

Comment 2: It would be nice for readers to explain why the author carried out at below zero degrees Celcius and how the experiment was performed.

Response: Thanks for your suggestion. Thermal stability is an important factor affecting drug activity and storage. According to the criteria for drugs in the Chinese Veterinary Medicine Canon, the relevant temperature sensitivity determination was carried out at different temperatures ranging from -20 °C to 60 °C. To determine the thermal stability, AMG was treated at different temperatures ranging from -20, 4, 37, 41, and 60°C for 1 h, respectively. Then the residual antibacterial activity was evaluated based on the broth microdilution method following the Clinical and Laboratory Standards Institute (CLSI) 2022 guidelines.

Comment 3: AMG possesses low MIC at 37 degrees Celcius which would be significant.

Response: Thanks for your suggestion. When performing the antibacterial test, we considered that a 1-fold difference in the MIC values was acceptable in the experimental error. So we considered AMG maintained the antibacterial activity after treatment under temperatures ranging from -20 °C to 37 °C for 1 h.

Comment 4: It would be necessary to explain the reason to conduct the activity from pH1.0 to pH6.0 at the temperature where the AMG showed low MIC towards S. aureus.

Response: Thank you for your suggestion. Since our main purpose was to evaluate the viability of AMG as an alternative strategy against Clostridial enteric infection, and AMG given orally would encounter low pH in the gastrointestinal tract (in this case, the pH value of chicken glandular stomach is 1.5-3.5), it would be necessary to determine whether AMG could have an effect on low pH condition.

Comment 5: How the author carried out the activity at dark also at below 0 degrees Celcius?

Response: Thanks for your suggestion. To keep the drug in dark conditions, AMG was pretreated with tin foil for 3 h, and the residual antimicrobial activity was evaluated against S. aureus. Similarly, AMG was pretreated for 1 h in the ice-water mixture, then the antimicrobial activity was determined.

Comment 6: Why author did not show AMG activity at physiological pH?

Response: Thanks for your suggestion. We have determined the activity of AMG at physiological pH in Figure R1C. Fig. R1C has been updated as Figure 3C in the revision.

Figure R1. Stability of AMG and AMG activity against C. perfringens isolates.

Comment 7: The author needs to improve the discussion part.

Response: Thanks for your suggestion. We have revised the discussion part.

Reviewer 2 Report

Good article

Minor English check

Accept

Author Response

Point-by-point response

Reviewer Comments:

Reviewer #2:

Good article, Minor English check, Accept

Response: Thank you for the insightful review. We have carefully and thoroughly proofread the manuscript to correct the grammar and typos.

Reviewer 3 Report

Herein, there were several major concerns that must be involved in the revised manuscript to meet the criteria for publication and carefully revise the English language in all manuscript.

Abstract:

The animal model should be described in detail (species, no, route the α-mangostin and it dose.

Results need more description.

Introduction:

Line 22: please replace it with another as it was the same as present in the abstract.

the repeated sentence as in lines 22 and 25.

Line 38: why the authors mentioned probiotics in this study?

Xanthones are described briefly in the introduction (please described the sources fungi or what and the mode of actions and effectiveness based on the previous studies in vivo and in vitro).

The animal model (species) must be described in the aim of the introduction.

Line 55: please remove it.

Results:

Line 133: broilers (what is the animal chickens, duck or turkey).

2.3. : where the material and methods of this part.

Material and methods:

 Line 253: please describe the details of the company.

The birds should be examined before experimental infection to be not suffer from higher clostridium count otherwise all results will be not representative.

Line 331: dose mg/kg ………..feed or body weight

Line 334: why the composition of diet was changed after 11 days of age.

Line 337: route of euthanasia with air (please add reference).

Line 335: The feed was withdrawn from day 14 to day 18 (four days too much, why????).

Line 342: The surgical procedures for what??.

Line 341: live or killed birds.

Author Response

Point-by-point response

Reviewer Comments:

Reviewer #3:

Herein, there were several major concerns that must be involved in the revised manuscript to meet the criteria for publication and carefully revise the English language in all manuscripts.

Response: We sincerely thank the reviewer for your professional and valuable comments. We have revised the manuscript carefully and thoroughly according to your suggestions.

Comment 1: Abstract: The animal model should be described in detail (species, no, route the α-mangosteen and its dose.

Response: Thank you for your suggestion. We have added more detail about the animal model in the abstract.

Comment 2: Results need more description.

Response: Thank you for the great suggestion. We have added more detail about the Results.

Comment 3: Introduction: Line 22: please replace it with another as it was the same as present in the abstract.

Response: We are very sorry for the repeated sentence. We have modified the relative sentences in the manuscript as follows.

“Intestinal infections caused by pathogenic bacteria are a serious health problem globally. The intestinal tract is an important reservoir for anaerobic pathogens, particularly Clostridium difficile and Clostridium perfringens. C. perfringens is the third common foodborn bacterial pathogen causing gas gangrene and food poisoning in human. In livestock and poultry industry, C. perfringens induces necrotic enteritis, a major concern and have led to severe economic losses.” (Line 23-28)

Comment 4: the repeated sentence as in lines 22 and 25.

Response: We have made corrections according to the Reviewer’s comments.

Comment 5: Line 38: why the authors mentioned probiotics in this study?

Response: We apologize for the confusion. We mentioned probiotics as one of the strategies employed to prevent clostridial enteric infections (Science Translational Medicine, 2020, 14, 657; Poultry Science, 2022, 101, 101590). However, the mechanism for probiotic therapy is unknown and complex in vivo, and probiotics are at risk of transmitting drug-resistant genes. Compared to probiotics, plant-derived natural products with antibacterial activity may be an effective strategy for preventing clostridial enteric infections. To make it more clearly, we have revised this sentence in the manuscript.

Comment 6: Xanthones are described briefly in the introduction (please described the sources fungi or what and the mode of actions and effectiveness based on the previous studies in vivo and in vitro).

Response: Thank you for the great suggestion. Many xanthones are phytochemicals found in plants in the families Bonnetiaceae, Clusiaceae, and Podostemaceae, especially in mangosteen peels (Botanical Journal of the Linnean Society, 2003, 141, 399). Additionally, based on the previous works, we find that xanthones are promising membrane-active antibacterial agents against bacterial infections and the ability of bacteria to acquire resistance is limited. We recently found that natural xanthones exhibit robust antibacterial activities against methicillin-resistant Staphylococcus aureus (MRSA) and vancomycin-resistant enterococci (VRE) (Journal of Agricultural and Food Chemistry, 2019, 67, 13195; Advanced Science, 2021, 8, e2100749; RSC Medicinal Chemistry, 2021, 13, 107). We have added descriptions of xanthones in the introduction.

Comment 7: The animal model (species) must be described in the aim of the introduction.

Response: Thanks for your suggestion. Necrotic enteritis is one of the essential enteric diseases in poultry and is a high cost to the industry worldwide. Clostridium perfringens, mainly type A and sometimes type C, causes necrotic enteritis in 2-to 5-week broilers. In this study, we evaluated the potential of plant-derived xanthones as an alternative strategy for the treatment of enteric infections in the broiler chickens’ necrotic enteritis model. We have added the relative sentences in the introduction.

Comment 8: Line 55: please remove it.

Response: Thanks for your suggestion. We have revised it.

Comment 9: Results: Line 133: broilers (what is the animal chickens, duck or turkey).

Response: Thank you for your reminder. We have revised the sentence to make it clear (Lines 174).

Comment 10: 2.3. : where the material and methods of this part.

Response: We thank the reviewer for the suggestion. We supplemented procedures in the manuscript accordingly (Line 444-455).

Comment 11: Material and methods: Line 253: Please describe the details of the company.

Response: We sincerely thank the reviewer for careful reading. The details of the company have been indicated at the beginning of this paragraph and the manuscript has been revised to avoid duplication.

Comment 12: The birds should be examined before experimental infection to be not suffer from higher clostridium count otherwise all results will be not representative.

Response: Thank you for the insightful comments. In this study, the bacterial load of C. perfringens was approximately 1×102-105 CFU/g in the cecum of healthy birds, which is much lower than the count of C. perfringens (1×107-109 CFU/g) after experimental infection, demonstrating the validity of the data (Figure S2A)

Figure S2. AMG is efficacious in the necrotizing enteritis model of clostridial enteric infections.

(A) Necrotizing enteritis model was established successfully. The broilers were euthanized after 5 days of persistent infection with C. perfringens. Bacterial loads (Log10 CFU of C. perfringens) in ileum and cecum were counted (n = 6).

Comment 13: Line 331: dose mg/kg ………..feed or body weight

Response: We sincerely thank the reviewer for careful reading. We are sorry for our carelessness. Based on your comments, we have made corrections to our dose unit, “mg/kg” was corrected as “mg/kg body weight”.

Comment 14: Line 334: why the composition of the diet was changed after 11 days of age?

Response: High dietary levels of animal byproducts (eg, fishmeal), wheat, or barley could predispose birds to the development of clinical necrotic enteritis symptoms following C. perfringens infection. Therefore, we changed the composition of the diet before the challenge period with C. perfringens culture.

Comment 15: Line 337: route of euthanasia with air (please add reference).

Response: Thanks for your suggestion. We have revised it and the corresponding references are indicated at the beginning of this paragraph, namely references 33 and 48 (Lines 428 and Lines 439).

Comment 16: Line 335: The feed was withdrawn from day 14 to day 18 (four days too much, why????).

Response: We are sorry for the misleading. Versatile toxins produced by C. perfringens are an important cause of intestinal damage in livestock and poultry. In order to make the toxins fully contact the intestinal epithelium and reduce the influence of intestinal contents, the feed was withdrawn each evening during the challenge period (day 14 to day 18).

Comment 17: Line 342: The surgical procedures for what??.

Response: We apologize for our careless mistake. In order to analyze the tissue distribution of AMG, the cecum contents were collected through surgical procedures. To make it clear, we have revised this sentence as follows.

“ The distributions of AMG in cecum were conducted according to a previously reported method.” (Lines 457-458)

Comment 18: Line 341: live or killed birds.

Response: We sincerely thank the reviewer for careful reading. We have revised this sentence as follows.

“C. perfringens was recovered from the intestines of killed birds by culturing on Tryptose Sulfite Cycloserine Agar Base.” (Lines 367-368).

Round 2

Reviewer 3 Report

Accept in the present form